# SARS-CoV-2 infection and antibody seroprevalence in routine surveillance patients, healthcare workers and general population in Kita region, Mali: an observational study 2020–2021

Dagmar Alber,[1] Fadima Cheick Haidara,[2] Juho Luoma,[3] Laura Adubra,[3] Per Ashorn,[3,4] Ulla Ashorn,[3] Henry Badji,[2] Elaine Cloutman-Green  ,[1] Fatoumata Diallo,[2] Rikhard Ihamuotila,[3] Nigel Klein,[1] Owen Martell,[5] Uma U Onwuchekwa,[2] Oumar Samaké,[2] Samba O Sow,[2] Awa Traore,[2] Kevin Wilson,[3] Camilla Ducker,[5] Yue-Mei Fan  [3]

DA and FCH are joint first authors.

CD and Y-MF are joint senior authors.

For numbered affiliations see end of article.

**Correspondence to**
Dr Yue-Mei Fan;
yuemei.fan@tuni.fi

## ABSTRACT

**Objective** To estimate the degree of SARS-CoV-2 transmission among healthcare workers (HCWs) and general population in Kita region of Mali.

**Design** Routine surveillance in 12 health facilities, HCWs serosurvey in five health facilities and community serosurvey in 16 villages in or near Kita town, Mali.

**Setting** Kita region, western Mali; local health centres around the central (regional) referral health centre.

**Participants** Patients in routine surveillance, HCWs in local health centres and community members of all ages in populations associated with study health centres.

**Main outcome measures** Seropositivity of ELISA test detecting SARS-CoV-2-specific total antibodies and real-time RT-PCR confirmed SARS-CoV-2 infection.

**Results** From 2392 routine surveillance samples, 68 (2.8%, 95% CI: 2.2% to 3.6%) tested positive for SARS-CoV-2 by RT-PCR. The monthly positivity rate was 0% in June–August 2020 and gradually increased to 6% by December 2020 and 6.2% by January 2021, then declined to 5.5%, 3.3%, 3.6% and 0.8% in February, March, April and May 2021, respectively. From 397 serum samples collected from 113 HCWs, 175 (44.1%, 95% CI: 39.1% to 49.1%) were positive for SARS-CoV-2 antibodies. The monthly seroprevalence was around 10% from September to November 2020 and increased to over 40% from December 2020 to May 2021. For community serosurvey in December 2020, overall seroprevalence of SARS-CoV-2 antibodies was 27.7%. The highest age-stratified seroprevalence was observed in participants aged 60–69 years (45.5%, 95% CI: 32.3% to 58.6%). The lowest was in children aged 0–9 years (14.0%, 95% CI: 7.4% to 20.6%).

**Conclusions** SARS-CoV-2 in rural Mali is much more widespread than assumed by national testing data and particularly in the older population and frontline HCWs. The observation is contrary to the widely expressed view, based on limited data, that COVID-19 infection rates were lower in 2020–2021 in West Africa than in other settings.

## STRENGTHS AND LIMITATIONS OF THIS STUDY

⇒ Our study was strengthened by the data available on participants from health facility clients, healthcare workers (HCWs) and the general population.

⇒ We investigated both active infection and seroprevalence for HCWs and the general population during the same time period.

⇒ First study of its kind in West Africa to provide routine surveillance data capturing.

⇒ Our study was performed in one region and therefore it only reflects the COVID-19 situation of the Kita region.

⇒ Extending active infection RT-PCR testing to non-symptomatic cases, both at the health facility and in the general population, would have provided more robust insights on infection rates as well as the level of asymptomatic cases.

## INTRODUCTION

The global COVID-19 pandemic, caused by the SARS-CoV-2, has had a devastating impact on the health of populations in all regions of the world, leading to over 5 million deaths and tens of millions of cases, also causing widespread and long-term social and economic damage.[1] Despite predictions made during the so-called 'first wave' (March–May 2020) that COVID-19 would spread widely and cause untold damage to communities in African countries, data during that period suggested that transmission, case and death rates have been lower in African settings than in other regions and continents.[2,3]

Whereas the reported numbers of SARS-CoV-2 infection have been low in large parts of Africa, there is some evidence that this may be due to undertesting or under-reporting. A

recent Zambian postmortem study suggested that many people who die have an undiagnosed SARS-CoV-2 infection.[4] Various studies have determined the seroprevalence in Africa.[5–13] A recent systematic review has estimated it to be approximately 22%, with highest seroprevalence in central Africa.[5] A recent study in Mali, conducted between November 2020 and June 2021, has reported a seroprevalence of 61.8% in healthcare workers (HCWs) who were in direct contact with patients with COVID-19 in Bamako Hospital.[12] However, little is known about the spread of SARS-CoV-2 in rural areas among HCWs and the general population.

The objective of our study was to characterise the epidemiology of COVID-19 epidemic in Kita region in western Mali, an area with very low rates of reported infections. To this end, we monitored time-trends in the proportion of positive SARS-CoV-2 test results among patients presenting with suspected SARS-CoV-2 infection in primary health centres over a period of 10 months and complemented the data with seropositivity for SARS-CoV-2 antibodies among personnel working in the same facilities during the same time frame. Furthermore, in the middle of the monitored time period, we conducted a population-based viral and serological survey, to establish a point-prevalence of acute infection and past exposure to it, among apparently healthy people of different ages.

## METHODS
### Study design and participants

The present study was designed to support the Malian Ministry of Health and Social Affairs' strategy to strengthen epidemiological surveillance and enhance the national response in Mali. The study was conducted in Kita region, Western Mali. The area is served by one referral health centre in Kita centre, while most health areas have a community health centre, offering a range of basic medical services with fee charged.

Our study targeted three groups—patients at a health facility, HCWs and apparently healthy community residents. We conducted surveillance for patients presenting at one referral health centre and 11 community health centres, over a period of 12 months. These patients presenting with symptoms of COVID-19 (respiratory symptoms and fever) were tested for active SARS-CoV-2 infection.

We conducted a HCWs serosurvey in five of the same health facilities over a period of 9 months and community serosurvey at one time point in 16 villages. Between September 2020 and May 2021, a total of 113 HCWs were approached in five health facilities and invited to participate in the HCWs SARS-CoV-2 serosurvey. The HCWs samples were from healthy nurses, doctors, office workers, laboratory personnel and other auxiliary staff who consented to participate in the study. Among them, 78.8% were nurses reflecting their high proportion among HCWs.

The community samples were selected on the basis of cluster sampling. We used enumeration areas as clusters and determined the number of households from each cluster based on its recorded population size. The actual households to be invited were selected by random sampling. The sample size was originally calculated to estimate the seroprevalence of SARS-CoV-2 antibodies with an absolute precision (margin of error) of 2.5% and assuming a baseline prevalence of 5%. Accounting for clustering effects of households within villages and of individuals within households, we estimated to require a minimum sample size of 3000. Assuming an average village population of 1446 (based on available regional statistics), average household size of 14 and average number of under-5 children per household of 3, we decided to visit 215 households to enrol the study sample. In December 2020, a total of 229 randomly selected households were visited in villages in or near the Kita centre and invited to participate in a SARS-CoV-2 community survey. Heads of all the invited households provided a verbal consent and a total of 905 participants presented in the house during the visit were included in the survey.

### Data and specimen collection

For the patient surveillance study, the study nurses filled in a patient screening form, a suspected infection enrolment form on exposures and symptoms and consenting form, followed by a medical examination and a flocked nasopharyngeal swab (NPS, flexible minitip, Copan) or oropharyngeal swab (OPS, flexible minitip, Copan; nasal and throat flocked swab, Dewei Medical Equipment Co., Ltd) specimen collection from those with a suspected SARS-CoV-2 infection, for example, patients experiencing fever, cough, shortness of breath and seeking medical advice. For the HCWs serosurveillance study, study nurses filled in a screening, consent and enrolment form, a suspected infection enrolment form on symptoms and exposure questionnaire. Four millilitre of venous blood sample (BD vacutainer, Cat. No. 369032) was collected monthly and an NPS or OPS was taken at enrolment and every 12 weeks thereafter. From each community survey participant, the nurse/data collector filled in a symptom and exposure questionnaire and collected a nasal swab specimen and a blood sample. A study nurse or a data collector filled out the symptom and exposure questionnaire using a REDCap electronic data capture application. The biological samples were collected by a doctor at the health centre and by a nurse in the community survey. The collected swab samples were transported at 2°C–8°C in 3 mL PBS/VTM from Kita to Bamako lab for analysis. The blood samples were centrifuged, and serum was aliquoted into cryovials at the Kita lab and batched samples were transported in dry shipper to Bamako for analysis.

### Nucleic acid extraction and real-time RT-PCR

RNA was extracted from sample swabs using a QIAamp Viral RNA Mini kit (Qiagen, Germany, Cat. No. 52906) according to the manufacturer's instructions.

Real-time RT-PCR assays for SARS-CoV-2 were carried out at the CVD-Mali Influenza laboratory, a participant in the WHO External Quality Assessment for the Detection of Influenza and SARS-CoV-2. Briefly, extracted RNA

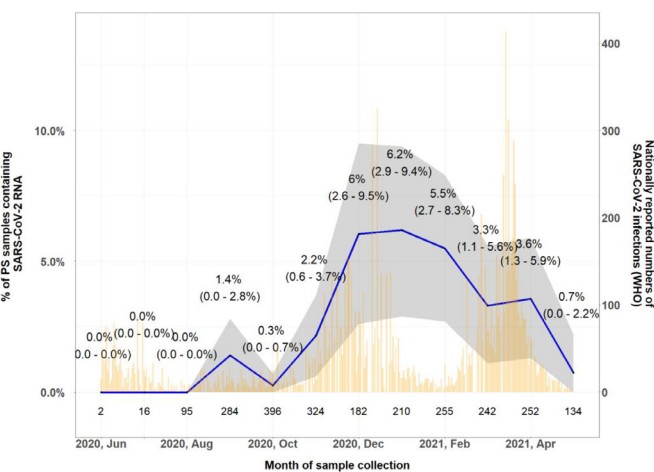

**Figure 1** Percentage of SARS-CoV-2 positive RT-PCR test by month in routine surveillance patients from health facilities. Grey area and values in parentheses indicate 95% CI from the proportion of SARS-CoV-2 RNA containing samples. Yellow lines indicate nationally reported numbers of SARS-CoV-2 infections (WHO https://covid19.who.int/region/afro/country/ml). Values above the x-axis indicate number of study samples in each month. PS, pharyngeal swab.

was amplified using the Superscript One-step RT-PCR kit (Invitrogen, Thermo Fischer Scientific, Cat No. 11732-088) targeting the RdRp gene (Corman VM, 2020) or the Quantabio qScript XLT One-Step RT-qPCR ToughMix targeting the N-gene (N2 primers) (CDC assay, https://www.fda.gov/media/134922/download).

All assays were validated using WHO RNA samples received as part of an external Quality Assessment Programme and a panel of known positive and negative samples from University College London, UK.

### Serological assays

A laboratory technician performed the Wantai anti-SARS-CoV-2 total antibody ELISA (target: RBD, qualitative assay detecting IgM and IgG, Beijing Wantai Biological Pharmacy Ent, Beijing, China) following the manufacturer's instructions. The sensitivity and specificity reported was 93% and 100%, respectively. Serum samples (100 μL) from each participant were tested as a singlet. Ten batches were performed for community survey samples, six batches for HCW samples and one for repeated samples.

Assay validation was carried out using the WHO international standard (NIBSC code: 20/136), the WHO International Reference Panel for anti-SARS-CoV-2 immunoglobulin (NIBSC code: 20/268) and an anti-SARS-CoV-2 Verification Panel for Serology Assays (NIBSC code: 20/B770). In addition, 100 pre-pandemic samples collected in 2012 were tested to assess the specificity of assay. All of these pre-pandemic serum samples were from adult women (mean age 26 years old, range 16–41).

Following the manufacturer's instructions, samples giving an absorbance equal to or greater than the cut-off value (A/CO>=1) were considered positive. Samples with A/CO ratio between 0.9 and 1.1 were considered borderline and were retested.

### Statistical analysis

We compared proportions, means and SD between groups by using Student's t-test for continuous variables and Fisher's exact test for proportions and $\chi^2$ test for global difference with p<0.05 indicating statistical significance. We calculated OR and 95% CIs with regression model. All statistical analyses were carried out using R V.3.4.4 and V.4.1.0.

### Patient and public involvement

Village chefs and health centre leaders were consulted before the study to ensure feasibility of the planned study procedures. Participants were not involved in the development of the research design, conduct of the research or preparation of the manuscript.

## RESULTS

Between June 2020 and May 2021, a total of 13 104 people presenting at outpatient clinics or using preventive services at 12 health facilities in or near the town of Kita were screened for exposure or symptoms of SARS-CoV-2 infection. Of these, 2393 were deemed to have a suspected SARS-CoV-2 infection and 2392 agreed to provide NPS or OPS samples for viral detection.

From all the tested samples from health centre patients, 68 (2.8%, 95% CI: 2.2% to 3.6%) gave a positive test result for SARS-CoV-2 by RT-PCR. Prior to August 2020, no SARS-CoV-2 RNA could be detected, infection rates steadily rose from November 2020 onwards, peaking at 6.2% (95% CI: 3.3% to 10.4%) in January 2021 and then gradually declined to 0.8% (95% CI: 0.02% to 4.1%) in May 2021 (figure 1). As shown in the figure, the increase in the proportion of positive samples followed a similar time trend in nationally reported numbers of SARS-CoV-2 infections and associated deaths.

Among the 113 invited staff members who all agreed to participate, 89 (78.8%) were nurses, 9 (7.9%) were doctors, 7 (6.2%) were administrative or laboratory technicians and 8 (7.1%) were auxiliary staff including housekeepers, cleaners and security staff.

Over the 9-month follow-up period, the participants provided a total of 397 blood samples, with a range of 1–7 samples per participant. Of these samples, 175 (44.1%, 95% CI: 39.1% to 49.1%) were positive for SARS-CoV-2 antibody by ELISA. By calendar month, the proportion of positive samples was approximately 10% until November 2020, increasing rapidly to 59.1% (95% CI: 38.5% to 79.6%) in January 2021 and remained relatively constant thereafter (figure 2). For different health professional groups, the proportion of SARS-CoV-2 antibody positive samples was highest among doctors (50.0%, 95% CI: 29.9% to 70.1%, 13/26 samples), followed by 49.8% (95% CI: 43.4% to 56.2%, 124/249) among nurses, 35.9% (95% CI: 21.2% to 52.8%, 14/39) among administrative staff and laboratory technicians and 15.4% (95% CI: 1.9% to 45.5%, 2/13) among auxiliary staff.

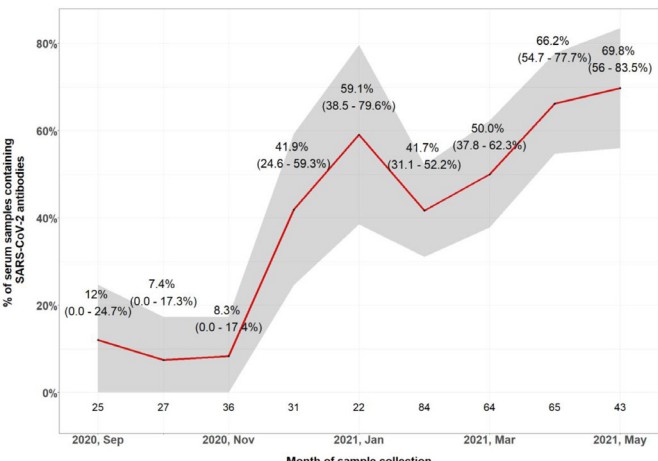

**Figure 2** Seroprevalence of SARS-CoV-2-specific total antibody by month in healthcare worker serosurvey. Grey area and values in parentheses indicate 95% CI from the proportion of SARS-CoV-2-specific antibody containing samples. Values above the x-axis indicate number of study samples in each month.

Of the 101 NPS or OPS samples collected from the health personnel, only one (1.0%) collected in November 2020 gave a positive test result for SARS-CoV-2 by RT-PCR.

Of the 905 community survey participants, 564 (62.3%) were women, 654 (72.3%) were at least 15 years old, 102 (11.3%) had respiratory symptoms in the preceding 14 days, 496 (55.4%) had attended a mass gathering in the preceding 3 months, none reported a contact with a COVID-19 patient and 21 (2.3%) had a fever. Eight hundred and eighty-two provided a blood sample for antibody analysis and 894 provided a nasal swab for

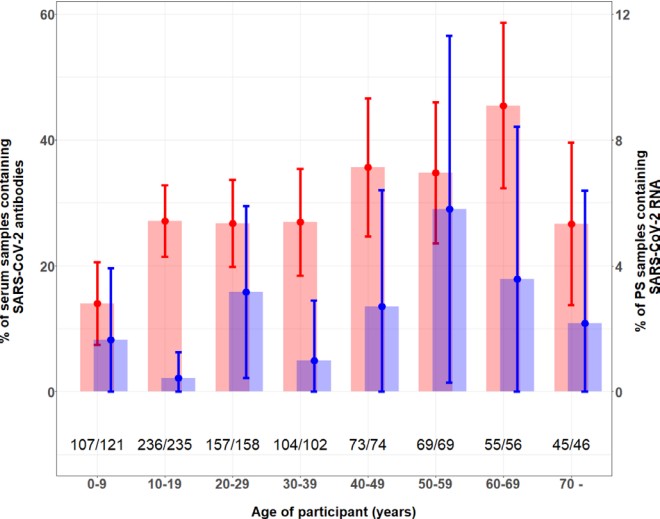

**Figure 3** Seroprevalence of SARS-CoV-2-specific total antibody and percentage of SARS-CoV-2 positive RT-PCR test by 10-year age interval group in community participants. Red bars indicate proportion (95% CI) of serum samples containing SARS-CoV-2-specific antibodies (primary y-axis), $\chi^2$ global age group difference, p=0.002; blue bars indicate proportion (95% CI) of PS samples containing SARS-CoV-2 RNA (secondary y-axis). PS, pharyngeal swab.

viral detection. Among all the participants, the overall seroprevalence was 27.7% (95% CI: 24.7% to 30.8%). The seroprevalence was lowest (14.0%, 95% CI: 7.4% to 20.6%) among 0–9 year-old and highest (45.5%, 95% CI: 32.3% to 58.6%) among 60–69 year-old participants (figure 3). Participants who had attended a mass gathering in the preceding 3 months had a higher seroprevalence than those who had not (odds ratio 1.48, 95% CI: 1.09 to 2.02). Participant sex, measured fever and recent history of fever or other respiratory symptoms were not associated with the presence of SARS-CoV-2 antibodies in the blood samples (table 1).

A total of 21 (2.4%, 95% CI: 1.5% to 3.6%) community survey participants had a positive RT-PCR result for SARS-CoV-2. The proportion of positive SARS-CoV-2 PCR tests was lowest (0.4%, 95% CI: 0% to 1.3%) among 10–19 year-old and highest (5.8%, 95% CI: 0.3% to 11.3%) among 50–59 year-old participants (figure 3).

Of the 100 banked serum samples collected before the COVID-19 pandemic (in year 2012), one gave a border-line positive result and 99 gave a negative test result in the Wantai SARS-CoV-2 antibody test.

## DISCUSSION

This study measured active and past COVID-19 infection in Malian HCWs and the general population through RT-PCR and seroprevalence testing. The results point towards significant underestimation of Mali's COVID-19 infection rates, as measured by current testing mechanisms. The study also provides insights into COVID-19 transmission patterns and risk factors. Extremely high infection rates were demonstrated by the study's serological testing. By December 2020, over a quarter (27.7%) of Kita's general population, and more than a third (41.9%) of HCWs had been infected. By January 2021, this had increased to 60% of HCWs.

National routine surveillance morbidity data for the study period (June 2020–May 2021) indicate that in June, transmission was at its first-wave peak, and that two later larger waves occurred, peaking in early December 2020 and April 2021.[14] Both the study's facility-based client/patient active infection investigation and HCWs seroprevalence findings indicate that Kita region's main transmission started later, in November 2020 and at the beginning it was very low, but increased gradually. Both study groups indicate a peak in December/January, similar to that seen nationally. The clients' active infection rate then showed a gradual decline to May rather than a second wave in March that was seen nationally.[14] This could be explained by containment measures reducing the emergence of a second peak in the general population.

A comparison of this study's seroprevalence and active infection rates show that seroprevalence is much higher than active cases in both the study's general population and in HCWs. The fact that this high seroprevalence did not translate into a similar expression of morbidity and resultant death, neither in our study nor in the national

**Table 1** Seroprevalence of SARS-CoV-2-specific total antibody by background characteristics

| | Sample size, n | Seropositive participants, n | Seroprevalence, % | OR (95% CI) |
|---|---|---|---|---|
| Participants, n | 875 | 242 | 27.7 | |
| **Sex** | | | | |
| Male | 330 | 98 | 29.7 | Ref |
| Female | 543 | 142 | 26.2 | 0.84 (0.62 to 1.14) |
| **Fever during past 14 days** | | | | |
| No fever | 832 | 229 | 27.5 | Ref |
| Fever | 41 | 11 | 26.8 | 0.97 (0.48 to 1.96) |
| **Fever (body temperature over 38°C)** | | | | |
| No fever | 852 | 238 | 27.9 | Ref |
| Fever | 21 | 2 | 9.5 | 0.27 (0.06 to 1.17) |
| **Respiratory symptoms in the past 14 days** | | | | |
| No symptoms | 773 | 216 | 27.9 | Ref |
| Any symptoms | 100 | 24 | 24.0 | 0.81 (0.50 to 1.32) |
| **Mass gatherings in the past 3 months** | | | | |
| No attendance | 370 | 86 | 23.0 | Ref |
| Attendance | 487 | 151 | 31.0 | 1.48 (1.09 to 2.02) |

surveillance system is an interesting area to investigate. This scenario has been documented quite widely in Africa.[15 16] This could be explained by a multitude of interlinking factors. High levels of asymptomatic cases, both for the general population and HCWs, have been documented in African as well as global studies.[11 17–21] As our study's active infection testing selected suspected cases of COVID-19 only, we could have missed a large proportion of asymptomatic cases. National surveillance would also have missed these cases as, first, they wouldn't present to a health centre and, second, even those that did would have been missed by diagnostic protocols.

Mali has gaps in demographic, disease morbidity and death surveillance, including for COVID-19. A recent Zambian study showed that nearly 20% of people who had died had a positive SARS-CoV-2 PCR test in a postmortem NPS sample. These people had not been tested before death, even though some of them had presented with COVID-19 symptoms.[4] A similar situation is likely present in Mali where diagnostic capacity is limited, at 7191 tests per 1 million people (compared with that of South Africa, 113 756) and where COVID-19 stigma exists.[22 23]

This study showed that 1.5 times more HCWs in Kita had been infected with SARS-CoV-2 by December 2020 compared with the general population. In fact, half of patient facing cadres such as doctors and nurses had been exposed to the virus and this was higher than laboratory, administrative (both 36%) and auxiliary staff (15.4%). Measures of increased risk of infection in HCWs versus the general population have been demonstrated in many global studies, with adequate personal protective equipment (PPE) provision and use being key to minimising this risk.[15 17 21 24] In resource-constrained settings such as Kita, key tools for protecting HCWs, along with early

disease detection, are often limited or absent in Africa.[25 26] Measures to improve protection for doctors and nurses, especially considering low levels of health staff availability in the country—in 2017 Mali had only six qualified providers per 10 000 inhabitants—would be pertinent.[27]

In the general population, risk factors included increased age and attendance at mass gatherings. Seroprevalence was lowest in the very young, and highest in 60–69 year-olds but again lower in the 70+ year olds. The possible reasons for lower seroprevalence in young children might be rapid waning of antibodies following mild symptoms and less exposures compared with adults. Mass gatherings are likely sources of infection given COVID-19's transmission dynamics and the difficulty of ensuring appropriate prevention control measures (mask wearing, social distancing, hand washing) at such events. Interestingly, neither fever (current or recent) nor other respiratory symptoms were seen to be significant risk factors, which again infers a high asymptomatic population.

The study's principal strength is that we included health facility clients, HCWs and the general population. We also investigated both active infection and seroprevalence for HCWs and the general population during the same time period. Study limitations include restriction to one region and therefore this study only reflects the COVID-19 situation of the Kita region. Extending active infection RT-PCR testing to non-symptomatic cases, both at the health facility and in the general population, would have provided more robust insights on infection rates as well as the level of asymptomatic cases. Widening the study to investigate exposure/transmission factors for study participants, such as PPE provision and use, social distancing, COVID-19 interventions and behaviour change would have allowed the study to explore risk factors to better

inform policy. When interpreting the RT-PCR results, possible sources of bias could have occurred within the sample collection regarding sample types (OPS/NPS), sample quality, transportation and temperature, and also in the use of three different RT-PCR assays. The latter is unlikely to have affected the results as all RT-PCR assays were validated in-house and by others.[28] Previous reports have suggested that some SARS-CoV-2 ELISA kits lack specificity when used on serum samples originating from Africa and in particular exhibit pre-existing cross-reactivity to SARS-CoV-2.[29 30] Our assay showed a 99% specificity when pre-pandemic samples were analysed and previous work has shown the Wantai assay to be highly sensitive (93%) and specific (100%).[31]

In conclusion, SARS-CoV-2 in rural Mali is much more widespread than assumed by national testing data and particularly in the older population and frontline health workers. The observation is contrary to the widely expressed view, based on limited data, that COVID-19 infection rates were lower in 2020–2021 in West Africa than in other settings.

Institutional preparedness as well as public risk assessment and personal protection capacity should be augmented, and behaviours in high-risk activities such as mass gatherings improved, if cessation of such events is not possible. Improving the understanding of COVID-19 in Mali through surveillance will enable the country to reduce the economic and social impacts of the disease. Further studies which provide accurate estimates of SARS-CoV-2 infection rates and seroconversion in the population would enable the development of a national transmission model, mapping of transmission hotspots, identification of national/regional risk factors and the prioritisation of equitable and effective responses. Such studies could also direct policy determinations in terms of disease prioritisation, ensuring that resources reallocated for COVID-19 control measures are used appropriately.

**Author affiliations**
[1]Great Ormond Street Institute of Child Health, University College London, London, UK
[2]Center for Vaccine Development–Mali, Bamako, Mali
[3]Center for Child, Adolescent and Maternal Health Research, Faculty of Medicine and Health Technology, Tampere University, Tampere, Finland
[4]Department of Pediatrics, Tampere University Hospital, Tampere, Finland
[5]Tro Da Ltd, London, UK

**Contributors** DA and FCH contributed equally and share first authorship. CD and Y-MF contributed equally and share last authorship. JL performed statistical analysis. PA, UA, NK, SOS, KW, CD and Y-MF initiated the project and preliminary design. DA, FCH, LA, HB, EC-G, FD, RI, OM, UUO, OS, AT developed the design and methods. DA, FCH, CD and Y-MF wrote the first draft of the manuscript. All authors contributed to the design of the study and interpretation of its results, and revised and approved the manuscript. PA and Y-MF are the guarantors and accept full responsibility for the work and the conduct of the study, had access to the data, and controlled the decision to publish. The corresponding author attests that all listed authors meet authorship criteria and that no others meeting the criteria have been omitted.

**Funding** The study was funded by Bill & Melinda Gates Foundation, grant number INV-017141. The funders had no role in study design, data collection and analysis, decision to publish or preparation of the manuscript.

**Competing interests** All authors have completed the ICMJE uniform disclosure form at www.icmje.org/coi_disclosure.pdf and declare: PA had financial support from Bill & Melinda Gates Foundation for the submitted work; no financial relationships with any organisations that might have an interest in the submitted work in the previous three years; no other relationships or activities that could appear to have influenced the submitted work.

**Patient and public involvement** Patients and/or the public were involved in the design, or conduct, or reporting, or dissemination plans of this research. Refer to the Methods section for further details.

**Patient consent for publication** Not applicable.

**Ethics approval** This study involves human participants and was approved by the Faculté de Médecine, Pharmacie et d'Odonto-Stomatologie (FMPOS) institutional review board in Mali and only participants who provided informed consent were enrolled in the study. All consents were requested verbally and recorded in the electronic data capture system. The minors' caregivers gave consent for their children to participate in the study. Participants gave informed consent to participate in the study before taking part.

**Provenance and peer review** Not commissioned; externally peer reviewed.

**Data availability statement** Data are available upon reasonable request. Deidentified participant data might be available on reasonable request to the corresponding author.

**ORCID iDs**
Elaine Cloutman-Green http://orcid.org/0000-0001-7534-3196
Yue-Mei Fan http://orcid.org/0000-0002-3241-6225

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
