## [Reviewer comments · BMJ Open]

ARTICLE DETAILS

TITLE (PROVISIONAL)	SARS-CoV-2 infection and antibody seroprevalence in routine surveillance patients, health care workers and general population in Kita region, Mali: an observational study 2020 - 2021
AUTHORS	Alber, Dagmar; Haidara, Fadima; Luoma, Juho; Adubra, Laura; Ashorn, Per; Ashorn, Ulla; Badji, Henry; Cloutman-Green, Elaine; Diallo, Fatoumata; Ihamuotila, Rikhard; Klein, Nigel; Martell, Owen; Onwuchekwa, Uma; Samaké, Oumar; Sow, Samba O.; Traore, Awa; Wilson, Kevin; Ducker, Camilla; Fan, Yuemei

VERSION 1 – REVIEW

REVIEWER	Gudina, Esayas Kebede Jimma University, Internal medicine
REVIEW RETURNED	03-Feb-2022

GENERAL COMMENTS	This work complements the limited but gradually increasing seroepidemiological studies of SARS-CoV-2 from Africa. The fact that the authors tried to investigate both active and previous infections among HCW and communities is highly commendable. The seroprevalence change over time in this study matches what has been reported from the continent. However, the manuscript can be presented in more attractive way than this. I suggest the authors to thoroughly review it by addressing the following issues and more. Abstract:  • I am not sure if this is according to the journal's guideline but the way the abstract was structured is not attractive • The objective was stated as "To estimate the degree of severe acute respiratory syndrome coronavirus-2 (SARS-CoV-2) transmission among health care workers (HCWs) and general population in a West African setting." However, only a few villages in western Mali were studied. This does not represent West Africa nor Mali. • In conclusion part, authors stated "Community spread of SARS-CoV-2 infection started in Kita in October-November 2020" but this is not substantiated by the findings. Introduction  • While 'Covid-19' is acceptable for general use, it is more appropriate scientifically to us COVID-19. So, correct Covid-19 as COVID-19 throughout the manuscript.
--

- I don't think reference 1 (Johns Hopkins Coronavirus Resource Center) has all the information stated in paragraph 1 of the introduction. Substantiate it with more appropriate citation.
- In general, the introduction is deficient and does not provide enough background information. Rather than just stating as '...very few community-based SARS-CoV-2 studies published so far from Africa...', authors should present summary of seroepidemiological studies from Africa. There serological studies from almost all regions of Africa. These are some:
 - <https://onlinelibrary.wiley.com/doi/10.1002/rmv.2271> - systematic review for Africa
 - 10.1016/j.puhe.2021.11.013 - Guinea-Bissau
 - <https://www.mdpi.com/1999-4915/14/1/102> - Mali
 - [https://www.ijidonline.com/article/S1201-9712\(21\)00596-8/fulltext](https://www.ijidonline.com/article/S1201-9712(21)00596-8/fulltext) - Nigeria
 - <https://gh.bmj.com/content/6/11/e007271> - Sierra Lionne
 - [https://www.thelancet.com/journals/eclinm/article/PIIS2589-5370\(21\)00452-1/fulltext](https://www.thelancet.com/journals/eclinm/article/PIIS2589-5370(21)00452-1/fulltext) - Zimbabwe
 - <https://pubmed.ncbi.nlm.nih.gov/33594353/> - South Africa
 - <https://jamanetwork.com/journals/jama/fullarticle/2784014> - Kenya
 - [https://doi.org/10.1016/S2214-109X\(21\)00386-7](https://doi.org/10.1016/S2214-109X(21)00386-7) – Ethiopia
 - <https://doi.org/10.1016/j.jiph.2021.09.011> - Egypt
- The last paragraph of the introduction, except the first statement, sounds like method and should be moved there.

Methods

- The study design was not clearly stated. Is this longitudinal cohort study, particularly, the serological study?
- Sampling technique should be placed here, not under statistical analysis part. Refer to my comments below.
- How the participants were selected (sampling technique) were described only in general terms. Authors should be more specific and provide the techniques and procedures used for selection of the three categories of participants.
- Both NPS and OPS were used for RT-PCR test. As the yield is not the same for both, would you please elaborate how you decided the type of specimen to collect?
- Under **Nucleic acid extraction and real time RT-PCR**, that much detail about RNA extraction and amplification is not relevant.
- **Serological assay** - *Wantai anti-SARS-Cov-2 total antibody ELISA* was used. It is important to provide more information about this assay. Which antibody does it detect (antiS/ antiNC/ or both; IgG/IgM or both)? What is the reported sensitivity and specificity?
- Statistical analysis – the first paragraph should be moved to Study design and participants section

Result:

- Paragraph 3, '*Between September 2020 and May 2021...*' first statement should be moved to methods part. There is an issue here that needs elaboration – the nurses made for nearly 80% of the participants. What is the rationale for this? If this is due to their proportion among HCW, it should be clearly elaborated in the methods part.

	 Paragraph 4, 'Over the 9-month follow-up period, the participants provided a total of 397 blood samples, with a range of 1-7 samples per participant. ...' what does this mean? If this is a longitudinal cohort, the authors should state how participants were recruited at each round. What is the seroincidence? Page 11, the paragraph containing 'In December 2020, a total of 229 randomly selected...', the first two statements of this paragraph should be moved to the methods part. Discussion:  Page 13, the first paragraph states that the study site (Kita area) has escaped the first wave. However, the data used in this study is not robust enough to support this claim. While it is possible, the assumption that the outbreak started after celebration of Prophet Mohamed's birthday on 29th October is not also substantiated by the data. The conclusion is not clear Figure  Figure 3 is less illustrative. It should be presented either as bar graph or line graph.
--	--

REVIEWER	Dhuria, Meera National Centre for Disease Control, Epidemiology
REVIEW RETURNED	06-Feb-2022

GENERAL COMMENTS	Title: timeline/ study period to be included in title of the study. Objectives: Please specify the objectives of the study more clearly. The authors seem have conducted three different studies (one on patients in healthcare facility, HCWs and Community) during different time frame and different Study designs, sampling techniques and testing strategies. Are all the three studies comparable? Kindly define the Objectives of the study specifically. Authors can consider classifying them into primary and secondary objectives. Study design: Since the serosurvey among HCWs is conducted over a period of 9 months, is it a longitudinal follow up study? Please provide details of sampling strategy for each sub group. How was an individual from a household selected? Also, specify the study period for each sub group. Data and specimen section: Kindly add inclusion and exclusion criteria for all the three sub-studies Statistical analysis: What is the basis of taking 5% base prevalence? How the design effect has been taken into consideration while estimating the sample size While estimating the seroprevalence, did the researchers took kit sensitivity and specificity into consideration. Results regarding test of significance have not been mentioned. Reason for calculating OR ANOVA test could have been run to statistically establish the difference in the mean seroprevalence among the different groups. Result: Authors can calculate Case to Infection Ratio or Infection to Case Ratio and cite similar articles for comparison. Discussion:
--

	Any reasons/ probable reasons can be cited to justify this finding of lower seroprevalance among paediatric population? Few studies show there is not much of a difference wrt serprevalance although number of detected cases are low among paediatric population. Please discuss Ethical considerations: to be included in the main write up of the manuscript. Pls specify if written informed consent was taken for drawing blood samples. Acknowledgement: Aditya Athotra, my colleague in Statistics Division has assisted in statistical review.
--	---

VERSION 1 – AUTHOR RESPONSE

Reviewer: 1
Dr. Esayas Kebede Gudina, Jimma University

Comments to the Author:

This work complements the limited but gradually increasing seroepidemiological studies of SARS-CoV-2 from Africa. The fact that the authors tried to investigate both active and previous infections among HCW and communities is highly commendable. The seroprevalance change over time in this study matches what has been reported from the continent.

Our response – We appreciate the reviewer’s comments.

However, the manuscript can be presented in more attractive way than this. I suggest the authors to thoroughly review it by addressing the following issues and more.

Abstract:

- *I am not sure if this is according to the journal’s guideline but the way the abstract was structured is not attractive*

Our response – We retained the abstract’s structure in keeping with the journal’s guidelines for authors.

- *The objective was stated as “To estimate the degree of severe acute respiratory syndrome coronavirus-2 (SARS-CoV-2) transmission among health care workers (HCWs) and general population in a West African setting.” However, only a few villages in western Mali were studied. This does not represent West Africa nor Mali.*

Our response – We have revised the objective part. The new version reads as below:

“To estimate the degree of severe acute respiratory syndrome coronavirus-2 (SARS-CoV-2) transmission among health care workers (HCWs) and general population in Kita region of Mali.”

- *In conclusion part, authors stated “Community spread of SARS-CoV-2 infection started in Kita in October-November 2020” but this is not substantiated by the findings.*

Our response – We have revised the conclusion part. The new version reads as below:

“SARS-CoV-2 in rural Mali is much more widespread than assumed by national testing data and particularly in the older population and frontline health workers. The observation is contrary to the widely expressed view, based on limited data, that

COVID-19 infection rates were lower in 2020-2021 in West Africa than in other settings.”

Introduction

- *While ‘Covid-19’ is acceptable for general use, it is more appropriate scientifically to us COVID-19. So, correct Covid-19 as COVID-19 throughout the manuscript.*

Our response – We have corrected all Covid-19 as COVID-19 throughout the manuscript as suggested.

- *I don’t think reference 1 (Johns Hopkins Coronavirus Resource Center) has all the information stated in paragraph 1 of the introduction. Substantiate it with more appropriate citation.*

Our response – We have now added the following references and revised manuscript accordingly, see below and in the revised manuscript.

1. World Health Organization. COVID-19 Weekly epidemiological update, 2021. Available: <https://www.who.int/publications/m/item/weekly-epidemiological-update-on-covid-19---7-december-2021>
2. Africa Centres for Disease Control and Prevention, *Latest updates on the COVID-19 crisis in Africa* (2021); <https://africacdc.org/covid-19>.
3. World Health Organization, *WHO coronavirus disease (COVID-19) dashboard* (2021); <https://covid19.who.int>.

- *In general, the introduction is deficient and does not provide enough background information. Rather than just stating as ‘...very few community-based SARS-CoV-2 studies published so far from Africa...’, authors should present summary of seroepidemiological studies from Africa. There serological studies from almost all regions of Africa. These are some:*

- o <https://onlinelibrary.wiley.com/doi/10.1002/rmv.2271> - systematic review for Africa
- o [10.1016/j.puhe.2021.11.013](https://doi.org/10.1016/j.puhe.2021.11.013) - Guinea-Bissau
- o <https://www.mdpi.com/1999-4915/14/1/102> - Mali
- o [https://www.ijidonline.com/article/S1201-9712\(21\)00596-8/fulltext](https://www.ijidonline.com/article/S1201-9712(21)00596-8/fulltext) - Nigeria
- o <https://gh.bmj.com/content/6/11/e007271> - Sierra Lionne
- o [https://www.thelancet.com/journals/eclinm/article/PIIS2589-5370\(21\)00452-1/fulltext](https://www.thelancet.com/journals/eclinm/article/PIIS2589-5370(21)00452-1/fulltext) - Zimbabwe
- o <https://pubmed.ncbi.nlm.nih.gov/33594353/> - South Africa
- o <https://jamanetwork.com/journals/jama/fullarticle/2784014> - Kenya
- o [https://doi.org/10.1016/S2214-109X\(21\)00386-7](https://doi.org/10.1016/S2214-109X(21)00386-7) – Ethiopia
- o <https://doi.org/10.1016/j.jiph.2021.09.011> - Egypt

Our response – We have added a short summary of seroepidemiological studies and references in the introduction part as suggested.

“Various studies have determined the seroprevalence in Africa. [5–13] A recent systemic review has estimated it to be approximately 22%, with highest seroprevalence in central Afrika.[5] A recent study in Mali, conducted between November 2020 and June 2021, has reported a seroprevalence of 61.8% in health care workers (HCWs) who were in direct contact with COVID-patients in Bamako Hospital.[12] However, little is known about the spread of SARS-CoV-2 in rural areas among HCWs and the general population.”

- *The last paragraph of the introduction, except the first statement, sounds like method and should be moved there.*

Our response – The last sentence describes the community where we investigated both active and previous infections. We have now revised the sentence.

“Furthermore, in the middle of the monitored time period, we conducted a population based viral and serological survey, to establish a point-prevalence of acute infection and past exposure to it, among apparently healthy people of different ages.”

Methods

- *The study design was not clearly stated. Is this longitudinal cohort study, particularly, the serological study?*

Our response – The study design was an observational cohort study which we have now also added in the title (see below).

“SARS-CoV-2 infection and antibody seroprevalence in routine surveillance patients, health care workers and general population in Kita region, Mali: an observational study 2020 - 2021”

As stated in the title, this study included three kinds of cohorts. In routine surveillance patients presenting in health centres, we tested active SARS-CoV-2 virus infection by PCR. In health care workers working in the same health centres and general population, both SARS-CoV-2 virus infection by PCR and serological ELISA tests were done.

- *Sampling technique should be placed here, not under statistical analysis part. Refer to my comments below.*

Our response – We have placed the sampling technique to the Method part as suggested.

- *How the participants were selected (sampling technique) were described only in general terms. Authors should be more specific and provide the techniques and procedures used for selection of the three categories of participants.*

Our response – We have added more information related to the sampling technique.

“We conducted surveillance for patients presenting at 1 referral health centre and 11 community health centres, over a period of 12 months. These patients presenting with symptoms of COVID-19 (respiratory symptoms and fever) were tested for active SARS-CoV-2 virus infection.

We conducted a HCWs serosurvey in five of the same health facilities over a period of 9 months and community serosurvey at one time point in 16 villages. Between September 2020 and May 2021, a total of 113 HCWs were approached in five health facilities and invited to participate in the HCWs SARS-CoV-2 serosurvey. The HCWs samples were from nurses, doctors, office workers, laboratory personnel and other auxiliary staff who consented to participate in the study. Among them, 78.8% were nurses reflecting their high proportion among HCWs.

The community samples were selected on the basis of cluster sampling. We used enumeration areas as clusters and determined the number of households from each cluster based on its recorded population size. The actual households to be invited were selected by random sampling. The sample size was originally calculated to estimate the seroprevalence of SARS-Cov-2 antibodies with an absolute precision (margin of error) of 2.5% and assuming a baseline prevalence of 5%. Accounting for clustering effects of households within villages and of individuals within households, we estimated to require a minimum sample size of 3000. Assuming an average village population of 1446 (based on available regional statistics), average household size of 14, and average number of under-5 children per household of 3, we decided to visit 215 households to enrol the study sample. In December 2020, a total of 229 randomly selected households were visited in villages in or near the Kita centre and invited to participate in a SARS-CoV-2 community survey. Heads of all the invited households provided a consent and a total of 905 participants were included in the survey.”

- *Both NPS and OPS were used for RT-PCR test. As the yield is not the same for both, would you please elaborate how you decided the type of specimen to collect?*

Our response – Our original plan was to collect NPS samples. However, as the swabs purchase and shipment were difficult at the beginning of COVID-19, some swabs delivered to the sites were a bit bigger, plus NPS sampling is a challenging procedure and OPS or nasal swab are easier to perform, we were not able to collect NPS for all study participants.

- *Under Nucleic acid extraction and real time RT-PCR, that much detail about RNA extraction and amplification is not relevant.*

Our response – We have deleted the details and only kept the necessary part as suggested.

- *Serological assay - Wantai anti-SARS-Cov-2 total antibody ELISA was used. It is important to provide more information about this assay. Which antibody does it detect (antiS/ antiNC/ or both; IgG/IgM or both)? What is the reported sensitivity and specificity?*

Our response – We have added more information about Wantai ELISA kits.

“A laboratory technician performed the Wantai anti-SARS-Cov-2 total antibody ELISA (target: RBD, qualitative assay detecting IgM and IgG., Beijing Wantai Biological Pharmacy Ent, Beijing, China) following the manufacturer’s instructions. The sensitivity and specificity reported was 93% and 100%, respectively.”

- *Statistical analysis – the first paragraph should be moved to Study design and participants section*

Our response – We have moved the first paragraph to Study design and participants section as suggested.

Result:

- *Paragraph 3, ‘Between September 2020 and May 2021...’ first statement should be moved to methods part. There is an issue here that needs elaboration – the nurses made for nearly 80% of the participants. What is the rationale for this? If this is due to their proportion among HCW, it should be clearly elaborated in the methods part.*

Our response – We have moved the first sentence of Paragraph 3 in Results part to Method part as suggested.

The nurses comprised of nearly 80% of HCW participants is due to their proportion among HCW. We have also added this in the methods part.

“Among them, 78.8% were nurses reflecting their high proportion among HCWs.”

- *Paragraph 4, ‘Over the 9-month follow-up period, the participants provided a total of 397 blood samples, with a range of 1-7 samples per participant. ...’ what does this mean? If this is a longitudinal cohort, the authors should state how participants were recruited at each round. What is the seroincidence?*

Our response – 113 HCWs consented to participate the study and we followed these participants at each round. Some participants gave verbal consent only in some rounds but not all and we were only able to collect a total of 397 blood samples. We then included this in our observational study to investigate the seroprevalence of SARS-CoV-2 antibodies among high exposure risk group (HCW) through the follow-up period.

- *Page 11, the paragraph containing ‘In December 2020, a total of 229 randomly selected...’, the first two statements of this paragraph should be moved to the methods part.*

Our response – We have moved these sentences to the Method part as suggested.

Discussion:

- *Page 13, the first paragraph states that the study site (Kita area) has escaped the first wave. However, the data used in this study is not robust enough to support this claim. While it is possible, the assumption that the outbreak started after celebration of Prophet Mohamed’s birthday on 29th October is not also substantiated by the data.*

Our response – We have revised this part in the Discussion.

“Both the study’s facility-based client/patient active infection investigation and HCWs seroprevalence findings indicate that Kita region’s main transmission started later, in November 2020 and at the beginning it was very low, but increased gradually. This suggests that the region escaped the earlier wave, with Covid-19 imported sometime in late October/early November, perhaps around the celebrations for the Prophet Mohamed’s birthday on 29th October.”

- *The conclusion is not clear*

Our response – We have revised this in the Discussion.

“In conclusion, SARS-CoV-2 in rural Mali is much more widespread than assumed by national testing data and particularly in the older population and frontline health workers. Community spread of SARS-CoV-2 infection started in Kita in October-November 2020, peaked in January 2021 and continued in May 2021. The observation is contrary to the widely expressed view, based on limited data, that COVID-19 infection rates were lower in 2020-2021 in West Africa than in other settings.”

Figure

- *Figure 3 is less illustrative. It should be presented either as bar graph or line graph.*

Our response – We have revised the Figure 3 as suggested.

Reviewer: 2
Dr. Meera Dhuria, National Centre for Disease Control

Comments to the Author:

Title: timeline/ study period to be included in title of the study.

Our response – We have included in the title as suggested. Revised text in title reads as:

“SARS-CoV-2 infection and antibody seroprevalence in routine surveillance patients, health care workers and general population in Kita region, Mali: an observational study 2020 - 2021”

Objectives: Please specify the objectives of the study more clearly. The authors seem have conducted three different studies (one on patients in healthcare facility, HCWs and Community) during different time frame and different Study designs, sampling techniques and testing strategies. Are all the three studies comparable? Kindly define the Objectives of the study specifically. Authors can consider classifying them into primary and secondary objectives.

Our response – We conducted our study in three cohorts – patients presenting in health centres, health care workers and general community members – to characterize the epidemiology of COVID-19 epidemic in Kita region in Mali. We have classified them as suggested.

“The objective of our study was to characterise the epidemiology of COVID-19 epidemic in Kita region in western Mali, an area with very low rates of reported infections. To this end, we monitored time-trends in the proportion of positive SARS-CoV-2 virus test results among patients presenting with suspected SARS-CoV-2 infection in primary health centres over a period of 10 months and complemented the data with seropositivity for SARS-CoV-2 antibodies among personnel working in the same facilities during the same time frame. Furthermore, in the middle of the monitored time period, we conducted a population based viral and serological survey, to establish a point-prevalence of acute infection and past exposure to it, among apparently healthy people of different ages.”

Study design: Since the serosurvey among HCWs is conducted over a period of 9 months, is it a longitudinal follow up study? Please provide details of sampling strategy for each sub group. How was an individual from a household selected? Also, specify the study period for each sub group.

Our response – The serosurvey among HCWs is to monitor the proportion of seropositivity and positive SARS-CoV-2 virus infection in the high exposure risk group over a period of 9 months. The details of sampling strategy and study period for each group are provided in the Methods – Study design and participants (see below too). All members of selected households were invited to enroll in the study.

“We conducted surveillance for patients presenting at 1 referral health centre and 11 community health centres, over a period of 12 months. These patients presenting with symptoms of COVID-19 (respiratory symptoms and fever) were tested for active SARS-CoV-2 virus infection.

We conducted a HCWs serosurvey in five of the same health facilities over a period of 9 months and community serosurvey at one time point in 16 villages. Between September 2020 and May 2021, a total of 113 HCWs were approached in five health facilities and invited to participate in the HCWs SARS-CoV-2 serosurvey. The HCWs samples were from nurses, doctors, office workers, laboratory personnel and other auxiliary staff who consented to participate in the study. Among them, 78.8% were nurses reflecting their high proportion among HCWs.

The community samples were selected on the basis of cluster sampling. We used enumeration areas as clusters and determined the number of households from each cluster based on its recorded population size. The actual households to be invited were selected by random sampling. The sample size was originally calculated to estimate the seroprevalence of SARS-Cov-2 antibodies with an absolute precision (margin of

error) of 2.5% and assuming a baseline prevalence of 5%. Accounting for clustering effects of households within villages and of individuals within households, we estimated to require a minimum sample size of 3000. Assuming an average village population of 1446 (based on available regional statistics), average household size of 14, and average number of under-5 children per household of 3, we decided to visit 215 households to enrol the study sample. In December 2020, a total of 229 randomly selected households were visited in villages in or near the Kita centre and invited to participate in a SARS-CoV-2 community survey. Heads of all the invited households provided a consent and a total of 905 participants were included in the survey.”

Data and specimen section: Kindly add inclusion and exclusion criteria for all the three sub-studies

Our response – These are added in Methods – Study design and participants.

“These patients presenting with symptoms of COVID-19 (respiratory symptoms and fever) were tested for active SARS-CoV-2 virus infection.

The HCWs samples were from healthy nurses, doctors, office workers, laboratory personnel and other auxiliary staff who consented to participate in the study.

Heads of all the invited households provided a verbal consent and a total of 905 participants presented in the house during the visit were included in the survey.”

Statistical analysis: What is the basis of taking 5% base prevalence? How the design effect has been taken into consideration while estimating the sample size

Our response – 5% was estimated from previous studies as prevalence of SARS-CoV-2 infection was low from published results when our study started.

The sample size estimate is calculated using relatively conservative assumptions (including clustering design effect of 2) so in practice slightly higher precision may be obtained. The sample size estimate was calculated analytically and validated, subject to the underlying assumptions, through simulation.

While estimating the seroprevalence, did the researchers took kit sensitivity and specificity into consideration.

Our response – We did not take kit sensitivity and specificity into consideration while estimating the seroprevalence as we didn't know which kit would be used at that time.

Results regarding test of significance have not been mentioned.

Our response – We have added this in Statistical analysis.

“We compared proportions, means and standard deviations (SD) between groups by using Student's *t* test for continuous variables and Fisher's exact test for proportions with $p < 0.05$ indicating statistical significance.”

Reason for calculating OR

ANOVA test could have been run to statistically establish the difference in the mean seroprevalence among the different groups.

Our response – We measure the association between different risk factors and presence/absence seropositivity, and OR can measure how strongly they are associated.

We thank the reviewer on the suggestion of considering the global hypothesis of difference between the groups. ANOVA requires certain assumption of normality that our dichotomously categorized outcome variable would violate. Although ANOVA may be robust against normality violations, we decided to conduct a Chi-squared test as an alternative.

“We compared proportions, means and standard deviations (SD) between groups by using Student's *t* test for continuous variables and Fisher's exact test for proportions

and Chi-squared test for global difference with $p < 0.05$ indicating a statistically significance.”

“Legend of Figure 3. Chi-squared global age group difference, $p=0.002$.”

Result: Authors can calculate Case to Infection Ratio or Infection to Case Ratio and cite similar articles for comparison.

Our response – We are unable to calculate Case to Infection Ratio as the data was not collected.

Discussion:

Any reasons/ probable reasons can be cited to justify this finding of lower seroprevalance among paediatric population? Few studies show there is not much of a difference wrt serprevalance although number of detected cases are low among paediatric population. Please discuss

Our response – We have added this in the Discussion.

“The possible reasons for lower seroprevalence in young children might be rapid waning of antibodies following mild symptoms and less exposures compared with adults.”

Ethical considerations: to be included in the main write up of the manuscript. Pls specify if written informed consent was taken for drawing blood samples.

Our response – Verbal consent has been requested from each HCW before participation in the study. For community serosurvey, heads of all the invited households provided a verbal consent and a total of 905 participants were included in the survey.

Acknowledgement: Aditya Athotra, my colleague in Statistics Division has assisted in statistical review.

Our response - We appreciate the comments.

VERSION 2 – REVIEW

REVIEWER	Dhuria, Meera National Centre for Disease Control, Epidemiology
REVIEW RETURNED	16-May-2022
GENERAL COMMENTS	The kit sensitivity and specificity is provided with the kit as literature. It is to be used to adjust prevalence. However, if it is not available, may be mentioned as limitation of the study. Also if the prevalence is high, the adjusted prevalence is not much affected by the kit sensitivity and specificity.